# Histone Deacetylase (HDAC) Inhibitors: A Promising Weapon to Tackle Therapy Resistance in Melanoma

**DOI:** 10.3390/ijms23073660

**Published:** 2022-03-27

**Authors:** Kostas Palamaris, Myrto Moutafi, Hariklia Gakiopoulou, Stamatios Theocharis

**Affiliations:** 1First Department of Pathology, Medical School, National and Kapodistrian University of Athens, 11527 Athens, Greece; kpalamaris@yahoomail.gr (K.P.); myrto.moutafi@yale.edu (M.M.); chgakiop@med.uoa.gr (H.G.); 2Department of Pathology, Yale University School of Medicine, New Haven, CT 06520-8023, USA

**Keywords:** melanoma, uveal melanoma, mucosal melanoma, HDAC inhibitors, therapy resistance, in vitro, in vivo models, clinical trials

## Abstract

Melanoma is an aggressive malignant tumor, arising more commonly on the skin, while it can also occur on mucosal surfaces and the uveal tract of the eye. In the context of the unresectable and metastatic cases that account for the vast majority of melanoma-related deaths, the currently available therapeutic options are of limited value. The exponentially increasing knowledge in the field of molecular biology has identified epigenetic reprogramming and more specifically histone deacetylation (HDAC), as a crucial regulator of melanoma progression and as a key driver in the emergence of drug resistance. A variety of HDAC inhibitors (HDACi) have been developed and evaluated in multiple solid and hematologic malignancies, showing promising results. In melanoma, various experimental models have elucidated a critical role of histone deacetylases in disease pathogenesis. They could, therefore, represent a promising novel therapeutic approach for advanced disease. A number of clinical trials assessing the efficacy of HDACi have already been completed, while a few more are in progress. Despite some early promising signs, a lot of work is required in the field of clinical studies, and larger patient cohorts are needed in order for more valid conclusions to be extracted, regarding the potential of HDACi as mainstream treatment options for melanoma.

## 1. Introduction

Melanoma is a malignant neoplasm, originating from melanocytes, a neural crest- cell derived lineage, that is responsible for the production and secretion of melanin pigment [1]. It occurs predominantly in the skin, either de novo, or within a benign or dysplastic nevus. Less commonly, melanoma arises in mucosal surfaces and within the uveal tract of the eye. It is the fourth most common cancer in the US, and the most lethal malignant neoplasm of the skin [2]. Clinicopathologically, cutaneous melanomas represent heterogenous neoplasms, classified into four main subtypes, with variations in their histological features and growth patterns: the nodular melanoma, the superficial spreading melanoma, lentigo maligna and acral melanoma. Uveal melanoma is the most frequent primary ocular malignancy and the second most predominant form of melanoma, while mucosal melanomas constitute a small percentage of the disease cases [3].

In early stage, surgical resection, with wide local excision, remains the standard-of-care first line treatment option for melanoma, with an over 90% 5-year overall survival rate (OS). In patients with advanced disease that has spread to regional lymph nodes or has offered metastases to other organs (stage III and IV), the need for reliable therapeutic strategies that can significantly prolong disease-free (DFS) and overall survival (OS) is urgent [4]. Recent developments in basic research have identified “signature” molecular alterations or distinguishing tumor traits, which can serve as targets for therapeutic intervention. Two hallmark features defining cutaneous melanomas are the constitutive activation of the MAPK intracellular cascade, through mutually exclusive gain-of-function mutations in *BRAF* and *NRAS* [5,6] and the high tumor mutational burden (TMB) [7]. The activation of the MAPK pathway is considered a global event, arising early in melanoma natural history and serves as the main driving force of melanoma oncogenesis, representing a promising therapeutic target. While RAS itself is considered as “undruggable”, inhibitors of both mutant BRAF and wild type MEK have been approved as first line treatments for locally-advanced and metastatic melanoma and have demonstrated improved tumor response rate and progression free survival (PFS) [8,9]. High TMB is also a property that can be targeted therapeutically, mainly because of its association with increased quantities of neo-epitopes and enhanced tumor immunogenicity [10]. This augmented capacity of melanoma to generate antigen specific CD8 T-cell tumoricidal responses has been exploited with the utilization of immunotherapeutic modalities, such as immune checkpoint inhibitors (ICIs), which have revolutionized patient care and demonstrated great clinical efficacy in a large percentage of melanoma cases [11]. Both PDL-1 (nivolumab, pembrolizumab) and CTLA-4 (ipilimumab) antibodies have been established as mainstream adjuvant therapeutic options in resected, stage III/IV tumors. Moreover, ongoing clinical trials suggest that their utilization as neo-adjuvant treatment protocols in resectable stage III melanomas could soon emerge as a standard-of-care therapeutic scheme for a significant fraction of patients. Mucosal and uveal melanomas deviate significantly from their cutaneous counterparts on a molecular level. Tumors arising in mucosal surfaces harbor driver mutations in *C-KIT*, *NF1* and *RAS* genes, while BRAF alterations are less frequently encountered [12]. On the other hand, uveal melanomas are defined by a completely distinctive and fascinating molecular profile and pathogenetic route, lacking almost all signature mutations of the other two melanoma subtypes. Indeed, their genetic background is characterized by overexpression of cyclin D1 and MDM2 and disruption of PI3K/Akt and MAPK pathways, via *PTEN* and *GNAQ11* mutations respectively [13]. A large proportion of uveal melanomas also bear loss-of-function mutations of *BRCA1-associated-protein-1 (BAP-1)* [13]. While immunotherapy, with ICIs, is regarded as a first line treatment for both mucosal and uveal melanoma, the lower frequency of *BRAF* mutations in these two subtypes means that BRAF inhibitors are of limited value [14,15]. Despite promising results, there is still a significant proportion of patients that are refractory to the available treatment options, or undergo recurrence after an initial regression. The enormous progress which has been made in deciphering the mechanisms that govern therapy resistance in melanoma has identified the principal role of epigenetic reprogramming in allowing tumor cells to evade elimination from therapeutic targeting [16].

The present review aims to summarize the accumulated knowledge regarding the potential role of a specific category of epigenetic drugs, namely histone deacetylase inhibitors (HDACi), in the effort to tackle therapy resistance and offer a novel treatment alternative to both cutaneous and non-cutaneous melanoma patients.

## 2. Therapy Resistance in Melanoma

Resistance of melanoma to BRAF inhibition can be elucidated in the context of two different models: the first model is intrinsic resistance, founded upon the Darwinian perspective of evolution, through multiple cycles of natural selection, while the second one is of Lamarckian origin and states that the drug itself prompts cell changes towards a more drug-tolerant state, that can persist across cell generations. Melanoma evolution proceeds through the emergence of multiple neoplastic clones, unequally distributed within the tumor mass, and defined by distinct molecular signatures. Some of the subclonal genetic perturbations, such as gain-of-function mutation in *NRAS*, *MEK1/MEK2* and *PI3K* or gene amplification of microphthalmia-associated transcription factor (MITF) offer alternative paths that allow tumor cells to overcome their addiction to constitutive activation of BRAF and bypass the blockade of this specific signal transduction route, endowing them with resistance to BRAF inhibitors [17,18]. Thus, upon drug exposure, these cell populations are enriched and reinforce therapy evasion. The Lamarckian model of acquired resistance could be mechanistically related to the arising in melanoma of two predominant cell populations, with distinct phenotypic states, termed as “invasive” (AXL high/MITF low) and “proliferative” (AXL low/MITF high) phenotypes, which harbor “signature” gene expression profiles, induced upon activation of distinct transcriptional programs [19,20]. The two phenotypes co-exist in tumors in various analogies and are also plastic in nature. Therefore, a tilt in the balance between the two transcriptional programs leads to reversible phenotypic interconversions. The invasive phenotype is associated with an inherent resistance to BRAF inhibition and tumors dominated by such cells show minimal response [21]. Moreover, upon drug exposure, tumor cells seem to reversibly undergo mesenchymal transition and deviate towards the invasive phenotypic state, as an adaptive response that allows them to evade eradication [22,23].

Immunotherapeutic approaches, based on ICIs, attempt to exploit the tumoricidal capacity of tumor infiltrating cytotoxic T-cells (TILS). The considerable clinical effect of immune-checkpoint inhibition in melanoma should be, at least partly, attributed to the natural tendency of this malignancy to prompt a robust immune reaction and a consequent dense lymphocytic infiltration. However, even though the development of a spontaneous host immune response and the generation of a T-cell inflamed microenvironment is a prerequisite, it is not a sufficient condition for the effective utilization of immunotherapeutic agents. Tumor cells, in a variety of malignancies, employ a diverse array of mechanisms to decrease their immunogenicity and render themselves “invisible” to the effector cells of adaptive immunity, or diminish their ability to interact with them in a stimulatory manner. Decreased expression of MHC-I [24] and upregulation of alternative immune-inhibitory molecules, such as TIM-3 and LAG-3 [25,26] are among the most prominent sources of resistance to ICIs. Interestingly, in melanoma, downregulation of MHC-I has been linked to the enrichment of a mesenchymal/invasive phenotypic profile in tumor cells, implying a coalescence of the mechanisms that confer resistance to both immunotherapy and BRAF inhibition [27].

Epigenetic deregulation modulates the transcriptional programs and influences the expression levels of various intracellular factors, that are either core elements of intracellular cascades or effectors of key cellular processes, such as proliferation, apoptosis and immunogenicity. The variability of genetic lesions that denote subclonal cell lineages emerging during tumor evolution are coupled to variations in the expression levels of myriad genes, that are in large part related to the heterogeneity in epigenetic regulatory mechanisms within different clones [28]. Moreover, epigenetic reprogramming is a driving force of phenotypic fluctuations that foster the adjustment of tumor cells to drug exposure and reinforce resistance to treatment [28]. Thus, the epigenetically-mediated reconstruction of tumor cell expression patterns is of principal importance for the acquisition of resistance to both BRAF inhibitors and checkpoint immunotherapies.

### 2.1. Epigenetic Alterations in Malignant Melanoma

Epigenetic modifications control gene expression, at a transcriptional, post-transcriptional or post-translational level, without altering the DNA nucleotides’ sequence. They engulf a diversity of mechanisms, such as DNA methylation, histone modifications and noncoding RNA mediated processes that elicit a re-adjustment in the activation state of various intracellular networks [29]. In tumor cells these modifications are imbalanced and deregulated, enabling them to acquire their malignancy traits [30].

Aberrant DNA methylation is a hallmark trait of malignant tumors, including melanoma, and the most well-studied epigenetic alteration in cancer [31,32]. Physiologically, it helps establish stable and heritable cell-type-specific gene expression patterns, through direct chemical modifications of nucleotides. Multiple malignancies are characterized by global hypermethylation and a consequential broad repression of tumor suppressor genes. In melanoma, the most frequently hypermethylated genes are cell-cycle checkpoint regulators, such as CDKN2A and negative regulators of PI3K/Akt and MAPK signaling pathways [31].

Noncoding RNAs include short and long noncoding RNA molecules that simultaneously control the expression patterns of multiple genes, at a post-transcriptional and post-translational level. Malignant tumors, in general are characterized by the aberrant expression of multiple noncoding RNA molecules, that enable cancer cells to acquire the hallmark properties of malignancy [33,34].

Histone modifications include dynamic alterations of chromatin architecture, that control the access of transcriptional machinery components to the condensed genomic DNA. They are classified into two distinct mechanisms: the ATP-dependent remodeling of nucleosome organization and the covalent modification of histone tails, by specific complexes, that catalyze the addition or removal of various chemical elements [35,36]. Histone covalent modifications include, among others, methylation, phosphorylation, acetylation and ubiquitination and create histone “marks” that represent transcriptional permissive or repressive enhancer and promoter landscapes. The regulatory mechanisms of these post-translational modifications and their contribution in delineating gene expression patterns are thoroughly reviewed elsewhere [35]. In melanoma both mechanisms of chromatin restructure are affected and deregulated. Somatic mutations within genes encoding components of chromatin remodeling complexes such as (p)BAF (i.e., SMARCC1, ARID1B, ARID2 and IDH1) [37,38] and Polycomb repressive complex [39] are detected at a relatively high frequency. Among covalent post-translational modifications, the one with the highest clinical significance is histone acetylation, which could serve as the target-of-interest of various treatments [40].

### 2.2. HDAC in Cancer

Histone acetylation controls gene expression and establishes permissive chromatin states by modulating access and binding of transcription machinery components to regulatory genomic regions. The covalent addition of acetyl groups to lysine residues of histone N-terminal tails loosens the chemical interactions between nucleosome subunits and DNA nucleotides and facilitates the decompensation of tightly packed chromatin, allowing access of transcription factors [41,42]. The acetylation of histones is a dynamic process, controlled by a fluctuating balance between the reversible activity of two enzyme families: histone acetyltransferases (HATs) and histone deacetylases (HDACs) [43]. HDACs are a chemically diverse group of enzymes, divided into four classes: class I, II (a, b), III and IV, according to their homology to their yeast analogs [44]. Class I HDACs include HDAC 1, 2, 3, and 8, which are characterized by ubiquitous expression and almost exclusively nuclear localization [44]. Class II HDACs display a tissue-specific pattern of expression and localize in both the nucleus and cytoplasm [44]. Besides their histone-mediated control of gene expression, this HDAC class regulates the activity of multiple intracellular components, by post-translational modifications. They are further divided into two subclasses: IIa (HDAC4, 5, 7, 9) and IIb (HDAC6, 10). The other two HDAC classes are Class III HDACs that include SIRT, and Class IV, which consists solely of HDAC11 [44].

HDAC levels are increased in multiple malignant tumors, including cutaneous and uveal melanoma, while they are also associated with multiple clinicopathological parameters and patients’ survival [45,46,47,48,49,50,51,52,53]. Their downstream targets encompass genes that control different cellular processes, such as proliferation, apoptosis, metabolism, and immunogenicity. The overexpression of HDACs in cancer and their critical role in regulating different aspects of tumor cells biology, suggests that they could serve as ideal therapeutic targets [54]. A number of HDAC inhibitors (HDACi) have been developed, and they are classified into categories, according to their chemical structure: (I) hydroxamic acids (hydroxamates); (II) short chain fatty (aliphatic) acids; (III) benzamides; (IV) cyclic tetrapeptides; and (V) sirtuin inhibitors including the pan-inhibitor nicotinamide and the specific SIRT1 and SIRT2 inhibitors sirtinol and cambinol, respectively [55]. They are widely used in both solid and hematologic malignancies, as well as in benign hematologic conditions and autoimmune diseases [56].

### 2.3. Cell Cycle—Apoptosis

Accumulated data from in vitro experiments in multiple cancer subtypes, including melanoma, suggest that HDAC inhibition shifts the balance of transformed cells towards diminished proliferation and enhanced activation of cell death. This antitumor effect depends predominantly on transcriptional activation of proapoptotic factors and cell-cycle checkpoint inhibitors, such as p21 and p53, along with suppression of cyclins, which drive progression through different phases of cell cycle [57,58,59]. HDACi also target and reinforce the activity p53/p21 axis at a post-translational level, in order to induce growth arrest. Various stress stimuli trigger p53 activation, by evoking a cascade of post-translational modification, such as acetylation and phosphorylation. Thus, deacetylation inhibition results in a generally higher acetylation status of p53 and maintenance of higher protein levels, accompanied by an enhanced transcriptional activity [60]. HDAC3 inhibition has been found to induce activation of the G2/M checkpoint, by modulating cyclin A acetylation and promoting its degradation [61].

### 2.4. Tumor Microenvironment—Angiogenesis

HDACi seems to provoke a variety of alterations, that enhance tumor cell immunogenicity and facilitate the stimulatory interactions between antigen presenting and adaptive immunity cells. Through upregulation of MHC-I/II [62,63,64], and co-stimulatory molecules CD80, CD86 [65], HDAC inhibition enables a more effective antigen presentation and priming of CD8 T-cells and make tumors more susceptible to cytotoxic cells tumoricidal activity. HDACi can also interfere with cancer angiogenesis by transcriptional repression of angiogenetic effectors, such as VEGFA, VHL and HIF-1a [66,67].

## 3. Pre-Clinical and Clinical Studies of HDAC-INHIBITORS in Melanoma

### 3.1. In Vitro Studies

In vitro studies have utilized multiple human and murine melanoma cell lines in order to assess the efficacy of HDACi and penetrate the mechanistic details of their tumor restraining potential. This approach has provided valuable insights regarding tumor cell-autonomous effects of deacetylase inhibition. Multifarious HDACi, with a variety of specificities, interfere with the vital processes of proliferation, apoptosis and migration in melanoma cells. In order to exert their multipronged tumor-suppressive activity, they simultaneously modulate the activation state of multiple heterotypic signaling cascades and block persistent oncogenic signals (Table 1, Figure 1).

Trichostatin (TSA) is an HDACi, that alleviates the mitogenic effect of Tle3 (transducing-like enhancer of split 3) in both human (HMV-II) and murine (B16) melanoma cell lines [68]. Tle3 acts as a co-factor that interferes with key cellular processes of proliferation and differentiation in a variety of tissues. Its integrative function in cellular physiology is in large part corroborated by the direct interaction and recruitment of HDACs in active genomic regions and suppression of several genes, including cell-cycle regulators such as cyclins D1 and A [68]. Thus, HDAC inhibition provokes a strong anti-proliferative effect through the inhibition of Tle3 activity. A similar proliferation-suppressing effect is also elicited by two additional HDACi: Apicidin and M344 [68].

Tenovin 6 targets sirtuins, a class of HDACs that integrate signals derived from different intracellular cascades and control different aspects of cell biology. In melanoma, sirtuin expression is indirectly downregulated by PI3K, via MITF repression, and their inhibition favors a proliferation arrest and an activation of programmed cell death, while it also compromises the cells’ invasive potential [69]. Administration of tenovin in human melanoma cell lines has been found to alter p53 acetylation status and consequently to increase the transcriptional levels of p53 controlled genes, including proapoptotic components Bax and Puma. Moreover, sirtuin inhibition amplifies the generation of intracellular reactive oxygen species (ROS) and prompts functional paucity and damage of mitochondria, while it also attenuates cells migratory capacity via repression of matrix metalloproteinases [69].

Compound (S)-8 is a hydroxamic-based HDACi, that has demonstrated great efficacy against human melanoma cell line A375. The multifaceted effects of the drug include cell cycle arrest, induction of caspase-dependent programmed cell death and diminution of motility via repression of matrix metalloproteinases [70].

AC-93253, a SIRT 2 inhibitor, and AR42, a phenyl-butyrate-based HDACi, are pivotal in reinstalling melanoma cell sensitivity to BRAF inhibition. The mechanistic background of their tumoricidal activity is founded upon targeting of collateral signaling pathways, which create a bypass route that allows tumor cells to evade inhibition of MAPK cascade [71,72]. AC-93253 directly modifies expression of EGFR, EPHRA2, EPHB1 and of additional downstream elements that carry out the transduction of oncogenic signals from the membrane receptors to the cell nucleus where they evoke the transcriptionally mediated responses that maintain tumor cells viability and growth [71]. AR42 main anti-tumor effect is the mediated activation of DNA damage response ATM signaling that incites a downregulation of many chaperone proteins (HSP70, HSP90) and leads to a concomitant downregulation of multiple tyrosine kinase receptors or intracellular signaling transduction components [72]. These HDACi could be core players in combination therapeutic regimens, along with BRAF targeting molecules and kinase inhibitors, such as dasatinib and pazopanib, which provide an initial hit to the mechanisms that fuel drug therapy evasion. Indeed, such combinations have already demonstrated very promising in vitro results, and could very soon enter the field of clinical trials as potent first-line schemes.

Valproic acid affects the activity of two of the four HDAC classes, namely HDAC I and II and incites apoptotic cell death in human melanoma cells [73], while it also sensitizes them to radiation [74].

Suberohydroxamic acid (SBHA) has also been shown to stimulate the intrinsic apoptosis pathway, through upregulation of the Bcl-2 family of proapoptotic proteins Bim, Bax, and Bak [75].

LBH589 (panobinostat) is an HDACi, with broad specificity and a capacity to sensitize intrinsically resistant melanoma cells to BRAF inhibition [76]. Indeed, the combination of BRAF inhibitor encorafenib and panobinostat induced caspase-dependent cell death in melanoma cell lines, including those initially resistant to BRAF inhibitors. The tumor-suppressive capacity of panobinostat derives from the blockade of complementary MAPK-independent signaling networks, such as downregulation of PI3K/Akt and c-myc [76].

Ginsenoside Rg3 has a specificity against HDAC3 while 6- and 8-Prenylnaringenin (6-PN, 8-PN) are considered as pan-HDACi. The tumor-restrictive effect of these HDACi is mediated predominantly by direct inhibition of the MAPK and PI3K/Akt pathways [77,78], while ginsenoside Rg3 has also been found to post-translationally upregulate p53, via modulation of the protein acetylation status [79].

Suberoylanilide hydroxamic acid (SAHA, Vorinostat) is an HDACi that induces growth arrest through activation of TGFb/smad4 signaling network [80]. In vitro experiments suggest that SAHA elicits an upregulation of activin A, an agonist of TGFb receptor in melanoma cell lines. Thus, a feed-forward self-perpetuating circuit is activated and leads to a subsequent intracellular phosphorylation sequence that ends up with the nuclear translocation of smad4 and induction of growth arrest [81].

LAQ824 (dacinostat) is an HDACi that restores melanoma cells vulnerability to inhibition of retinoic acid signaling system [80]. Retinoic acid receptor (RARb2) is downregulated in melanoma, through an epigenetic mechanism, based on the establishment of repressive histone modification patterns. Thus, HDAC inhibition with dacinostat reinstalls a transcriptionally permissive state at RARb2 regulatory elements and restores the dependence of tumor cells to the retinoic acid tumor suppressive effect, an alteration that can be exploited therapeutically. The simultaneous inhibition of HDAC and retinoic acid agonism suppresses tumor cell growth and represents a novel and promising therapeutic approach for malignant melanoma [80].

MC1568 and MC1575 are class II-specific HDACi, that suppress the expression levels of IL-8 and c-Jun, via alterations of the acetylation status at the genes’ regulatory regions. IL-8 and c-Jun formulate a self-perpetuating intracellular circuit and through reciprocal activating interaction provide strong proliferative and antiapoptotic signals [82]. Thus, their diminished activity, through modified epigenetic landscapes, fosters a robust anti-tumor effect in human melanoma cell lines. HDAC inhibition interferes with different steps of this signaling network, by abrogating the access and binding of transcription machinery components at the two genes’ regulatory elements. C-Jun expression is impaired through inhibition of RNA polymerase II and TFIIB recruitment at the gene promoter, and the inhibitory effect is exponentially increased through impairment of c-Jun binding at IL-8 regulatory elements [82].

Quisinostat is an HDACi, that has demonstrated immunomodulatory activity on uveal melanoma cell lines, by enhancing antigen presentation capacity of tumor cells, via upregulation of MHC-I expression and cell-surface levels [83].

### 3.2. In Vivo Studies

The tumor-restraining effects of many HDACi have been substantiated in murine xenograft models, generated by cutaneous/subcutaneous inoculation of both human and murine melanoma cell lines in syngeneic mice. The employment of such in vivo approaches offers two main advantages: the verification of the tumor-suppressive activity at an organism level and the evaluation of the drug effect on interactions between tumor cells and components of the tumor microenvironment (Table 1, Figure 1 and Figure 2).

Tumor cell-autonomous effects: A number of HDACi have demonstrated promising results when evaluated in different in vivo models. TSA impaired the progression of xenograft tumors generated by B16 melanoma cell line subcutaneous injections in a similar manner as in vitro experiments mediated by Tle3 downregulation [68]. The treatment of melanoma-bearing mice with a combination of LAQ824 (dacinostat) and cis-retinoic acid, an RARb2 agonist, showed promising results in impeding tumor growth, via cell cycle arrest and the induction of apoptosis [80]. Ricolinostat (ACY-1215) is a selective HDAC6 inhibitor that induces cell-cycle arrest and activates apoptotic cell death in melanoma cells, while it has demonstrated significant capacity in inhibiting xenograft tumors outgrowth [84].

An emerging therapeutic strategy is based on the design and construction of hybrid pharmacologic agents, with multifaceted pharmacodynamics, that allow them to target simultaneously different cellular complexes and enzymes. Corin is a recently designed compound with dual specificity against HDAC and LSD1, that has been assessed in xenograft models of the SK-MEL-5 cell line [85]. This hybrid agent has demonstrated impressive potential in compromising tumor growth by inducing upregulation of tumor suppressor genes p21 and mxd1 [86].

Murine xenograft models are also an invaluable tool for the evaluation of combination therapeutic schemes, consisting of an HDACi and a chemotherapeutic drug or an agent targeting tyrosine kinase receptors of collateral oncogenic pathways.

The inhibition of HDAC1/2/3 by valproate or entinostat sensitized melanoma cells to the chemotherapeutic agent temozolomide and impeded melanoma xenografts outgrowth [85]. This tumoricidal effect is partly mediated by a compromise of the DNA double-strand-breaks repair mechanism, through downregulation of two homologous recombination complex components: RAD1, FANCD2. Consequently, the two class I HDACi demonstrated synergistic activity with PARP-1 inhibitor Olaparib [85]. Another selective class I and II HDACi, AR42, showcased synergistic tumor-suppressive potential with multi-kinase inhibitor pazopanib.

Tumor microenvironment effects: Melanoma progression is accompanied by a reconstruction of adjacent stroma and the recruitment of heterogenous leukocyte populations, that maintain a tumor-associated inflammatory reaction. An elegant network of cytokines, chemokines and growth factors tightly regulate the accumulation and leaning of myeloid cellular components towards immunosuppressive phenotypes, such as myeloid-derived suppressor cells (MDSCs) and tumor-associated macrophages (TAMs). These tumor-permissive constituents of the immune system engage in intricate interactions with neoplastic cells to directly promote tumor growth. Moreover, they secrete a large breadth of anti-inflammatory cytokines and other molecules, such as TGF-b, Interleukin-10 (IL-10), Interleukin-6 (IL-6) and Arginase 1 (ARG1). Hence, they circumvent the effective priming of the adaptive immunity compartment, via compromising effector CD8 T-cells recruitment to the tumor site or by maintaining them in a state of functional paucity. They therefore create an immunologically “cold” or immune-tolerant microenvironment that represents an impenetrable obstacle for the successful utilization of immunotherapeutic modalities. HDACi deploy a diverse array of mechanisms in order to restructure the tumor-immune microenvironment and rewire T-cells, in favor of a more robust adaptive immune response (Table 1, Figure 2) [87].

CCL-2 is an HDAC-regulated chemokine, with a pivotal role in facilitating the generation of a tumor-permissive and immune-suppressive microenvironment, via the accruement of myeloid immunosuppressive cell populations. Thus, the administration of HDACi SAHA in melanoma-bearing mice diminished the CCL2-driven accumulation of host MDSCs at tumor site, allowing effective generation of an antitumor immune response [88].

Valproic acid, administered in a similar model of xenograft tumor-bearing mice, exerted an immunomodulatory effect, by inducing a phenotypic switch of MDSCs at the TME and dictating their change from an immunosuppressive towards an immune-stimulatory and tumor-suppressive state [89]. This phenotypic conversion is mediated via activation of IRF1/IRF8 axis, that incites a downregulation of anti-inflammatory cytokines and mediators, such as Interleukin-10 (IL-10), Interleukin 6 (IL-6) and Arginase-1 (ARG1).

In a similar manner, Nexturastat A, a selective HDAC6 inhibitor, rewired TAMs towards a M1 phenotype, via transcriptionally mediated repression of anti-inflammatory paracrine factors, including TGF-b, IL-10 and ARG1. In contrast, M1 macrophages secrete a variety of proinflammatory growth factors and mediators that facilitate the effective stimulation and priming of effector CD8 T-cells [90].

Therefore, the exclusion of MDSCs and TAMs from the TME and their functional impairment enabled the transition from a “cold”, tumor-permissive microenvironment, to a “hot” immunologically active one and incited the generation of a productive adaptive immune response. The increased levels of CD8 T-cell infiltration potentiated the tumor-restrictive capacity of anti-PDL-1 treatment [89,90].

Multiple experimental studies have utilized melanoma mouse models, generated by B16 cell line, in order to evaluate the efficacy of HDAC inhibition, combined with vaccines or other immunotherapeutic approaches in containing tumor outgrowth. A combination therapy, composed of HDACi depsipeptide and BET inhibitor IBET151 resulted in enhanced cellular and humoral immune response and augmented the therapeutic efficacy of an OVA vaccine, in a B16 xenograft mouse model [91].

Depsipeptide has also demonstrated noteworthy potential in magnifying the tumor-restrictive capacity of antigen-specific adoptive T-cell therapy in melanoma-bearing mice, by exerting pleiotropic effects in both neoplastic and immune cells. The most prominent tumor cell-autonomous functions include the enhanced expression of MHC-antigen complexes and the sensitization of neoplastic cells to Fas-FasL mediated apoptotic cell death. In parallel HDAC inhibition amplified T-cells cytotoxic capability, via upregulation of effector molecules, such as Granzyme B [92,93].

HDACi LBH589 exerts multifaceted tumor-containing effects and renders melanoma cells more susceptible to elimination by ICIs and antigen-specific adoptive T-cell immunotherapy. Mechanistic studies in B16 xenograft tumors, revealed a direct role of HDAC inhibition in loosening chromatin structure in PDL-1/PDL-2 regulatory elements and upregulating their expression. Consequently, therapeutic regimens consisting of PDL-1/PDL-2 and HDAC inhibitors significantly reduced tumor burden and prolonged mice survival. Moreover LBH589 generates a microenvironment highly rich in pro-inflammatory cytokines and induces higher levels of IL-2 receptors (CD25) and OX-40 receptors on T-cells, propagating the antitumor effect of adoptive T-cell immunotherapy [94,95].

## 4. Clinical Trials

A multiplicity of phase I and II clinical trials have been completed or are still in progress, in order to evaluate toxicity and tolerability of HDAC inhibitors and assess their efficacy in melanoma patients, predominantly in the context of combination therapeutic schemes (Table 2).

### 4.1. Phase I Clinical Trials

Multiple phase I clinical trials have evaluated the clinical efficacy of two HDACi, vorinostat and panobinostat, in the setting of combination treatments, along with traditional chemotherapeutic agents, BRAF inhibitors, proteasome inhibitors or ICIs. Vorinostat and proteasome inhibitor marizomib, have demonstrated synergistic anti-tumor activity, exerted via the targeting of complementary intracellular networks that sum up to the amplification of cellular stress. A phase I clinical trial (NCT00667082) tested the efficacy of this combination regimen in heterogenous malignancies, including melanoma, non-small cell lung cancer and pancreatic ductal adenocarcinoma [98]. The most encouraging data extracted from this study concerned melanoma patients, as over 50% of them displayed disease stabilization, after two treatment cycles [98]. Mechanistic data also suggest that vorinostat potentiates the cytotoxic potential of topoisomerase inhibitor doxorubicin, by inducing decompensation of tightly packed genomic DNA and facilitating access and binding of topo II inhibitors to their substrate DNA. Thus, a phase I study (NCT00331955) was conducted, in order to assess whether this synergistic activity can be exploited therapeutically, in a variety of tumors [99]. A total of 32 patients, with a diverse array of malignancies, including six with melanoma, were recruited to the study. Two of the six melanoma patients met the criteria for disease stabilization for more than 8 months [99].

Another phase I clinical trial (NCT01065467) evaluated the antitumor activity of panobinostat in unresectable stage III or IV melanoma [100]. A total of 16 patients were enrolled in the study and were segregated into two arms. Half of the six patients of the first arm (Arm A) experienced severe toxicity, with significant thrombocytopenia. As a result, a second arm (Arm B) was created with 10 additional patients, administered doses at a lower frequency, in order to diminish side-effects. The aggregated efficacy analysis for both arms revealed a lack of partial or complete response in the total of 16 patients, while a total of four patients, two in each arm, demonstrated disease stabilization [100]. The conclusion extracted from this study is therefore against the potential use of panobinostat monotherapy for melanoma [100]. A second phase I clinical trial was conducted to assess the tolerability and efficacy of panobinostat in combination with a second epigenetic drug (decitabine: a DNA methyltransferase inhibitor) and a traditional chemotherapeutic drug (temozolomide: alkylating agent) (NCT00925132) [104]. The study involved 20 patients, with stage III or IV cutaneous, ocular or mucosal melanoma. A single patient, with mucosal melanoma, showed complete response for 8 months, while five other patients displayed disease stabilization [104].

Immunotherapy with ipilimumab, an anti-CTLA-4 antibody, is among the established therapeutic agents in metastatic melanoma. The combination of ipilimumab with the HDACi panobinostat was administered in patients with unresectable stage III or stage IV melanoma (NCT02032810). The response rate did not appear to be affected by the use of panobinostat [97].

A phase I dose-escalation trial, aiming to evaluate the efficacy of quisinostat (JNJ-26481585) in a broad spectrum of malignancies has also been conducted. Among the 92 patients enrolled to the study, 22 had melanoma. Quisinostat monotherapy induced complete response to just one melanoma patient, while partial response was encountered in two patients [101].

### 4.2. Phase II Clinical Trials

A phase II clinical trial (NCT00121225) tested vorinostat in a cohort of 32 patients with advanced cutaneous or ocular melanoma. Two patients experienced partial response and 16 demonstrated disease stabilization. Patients with stable disease or partial response had a median PFS of 5 months and the study was terminated, as it did not meet its primary endpoint criteria [108]. Moreover, vorinostat monotherapy was associated with significant toxicity in a number of patients who experienced severe side-effects, such as nausea, lymphopenia and hyperglycemia [108].

ENCORE-601 study is a phase II clinical study (NCT02437136), that assessed the combination of HDACi entinostat with pembrolizumab (anti PD-1) in 53 patients with unresectable or metastatic melanoma, who had experienced disease progression during or after pembrolizumab monotherapy. The results were promising, as the addition of HDACi demonstrated significant clinical activity [102].

The efficacy of the entinostat-pembrolizumab combination has also recently been evaluated in advanced or metastatic uveal melanoma. PEMDAC clinical trial is a phase II study (NCT02697630), that enrolled 29 patients bearing uveal melanoma. Eight patients demonstrated disease stabilization, while four others experienced partial response [106,107].

Another phase II trial (NCT00185302) evaluated the efficacy of single-agent treatment with HDACi MS-275 in 28 patients with unresectable or metastatic melanoma, refractory to at least one previous systemic therapy. MS-275 monotherapy induced disease stabilization in seven patients [105].

Another interesting combination therapeutic scheme, consisting of HDACi valproic acid and topoisomerase I inhibitor karenitecin, was evaluated in a phase I/II trial (NCT00358319). The double-agent regimen evoked disease stabilization, for up to 50 weeks, in 13 of 39 patients with AJCC stage IV melanoma [103]. The clinical efficacy of valproate, in the setting of combination therapeutic schemes was also tested in a phase I/II trial, along with immunochemotherapy with dacarbazine and interferon a. The combination did not have superior results to the standard of care treatment and there were considerations about the efficacy of VPA administration schedule [109].

Besides the already completed studies, there are a number of ongoing clinical trials aiming to test the efficacy of HDACi either as monotherapeutic approach, or in combination schemes, predominantly with immune checkpoint inhibitors. Two studies inclined to evaluate the potential clinical value of HDACi single-agent therapies are currently in progress, utilizing vorinostat [96] in patients with advanced or metastatic disease, or in patients with tumors refractory to previously administered systemic therapy. The synergistic effect of HDAC inhibition and immune checkpoint blockade is also under evaluation in phase I and II clinical trials, utilizing combination schemes, consisting of HDACi, along with pembrolizumab or nivolumab. Another novel therapeutic strategy, currently under assessment in a phase II clinical trial is based on the simultaneous inhibition of HDACs and MEK1/2 via belinostat and binimetinib, respectively.

## 5. HDACi Limitations–Adverse Effects

Despite the promising results from clinical evaluation of some HDACi, clinical studies have also unraveled significant limitations, concerning their effective deployment as a standard-of-care treatment approach. The majority of tested molecules seem to offer a clinical benefit in a minority of enrolled patients, in the form of partial response or disease stabilization, while almost no case of complete response has been detected. Moreover, within a relatively short period of time, the majority of responders undergo disease progression. Their clinical efficacy is also restricted to the context of combinational therapeutic schemes, while their utilization as monotherapy has failed to affect disease progression. A number of HDACi have also been linked to the occurrence of Grade 3 (severe) and Grade 4 (life-threatening) adverse effects in a proportion of enrolled patients. The most commonly encountered side effect is bone marrow paucity, in the form of hemopoiesis impairment. Multiple hemopoietic lineages, including lymphoid, erythrocytic, granulocytic and megakaryocytic are simultaneously affected by vorinostat and panobinostat toxicity, resulting in lymphopenia, anemia, neutropenia and thrombocytopenia, respectively. Vorinostat is also linked to a disruption of glucose homeostasis, leading to hyperglycemia and associated symptoms, while quisinostat has demonstrated a different profile of toxicity, related to gastrointestinal disturbances, such as nausea and vomiting.

## 6. Conclusions

Epigenetic rewiring integrates signals from multiple heterotypic intracellular networks and is therefore a key driver of melanoma resistance to the two mainstream therapeutic approaches: immunotherapy and *BRAF* inhibition. Even though epigenetics encompasses a broad spectrum of mechanisms, governing regulation of gene expression at different levels, histone deacetylation is considered the most promising to be targeted therapeutically. Indeed, a wide range of HDACi, defined by a diversity of chemical structure and target specificities have been developed and evaluated in multiple malignancies. In melanoma, both in vitro and in vivo experimental models and clinical trials have unraveled the pivotal role of HDACi in improving response and prolonging patient survival. Experimental evidence has provided mechanistic details about HDAC’s functional role in the melanoma evolutionary trajectory that strongly supported a potential benefit from a potential application of HDACi as part of combination therapeutic schemes. Indeed, multiple clinical studies have deployed HDACi in the setting of combination regimens, along with immunotherapeutic agents, almost exclusively in locally advanced, metastatic or refractory disease. Additional therapeutic regimens include proteasome and MEK1/2 inhibitors, as well as classic chemotherapeutic agents such as topoisomerase inhibitors. While some clinical studies have demonstrated significant results, in the form of disease stabilization or partial response, there is still much room for improvement. The majority of the completed studies involved relatively small patient samples and the number of HDACi tested was restricted compared to the abundance of molecules available. It is therefore of utmost importance to conduct large scale in vitro screening of potential HDACi in order to identify more capable candidate molecules, which could further proceed to in vivo assessment and, if appropriate, to clinical trials. Moreover, a number of already ongoing clinical trials will help in the accumulation of more data regarding the potential of HDAC inhibition as a future mainstream therapeutic approach in melanoma. However, the validity of the results extracted from these studies needs to be enhanced by larger patient cohorts.

## Figures and Tables

**Figure 1 ijms-23-03660-f001:**
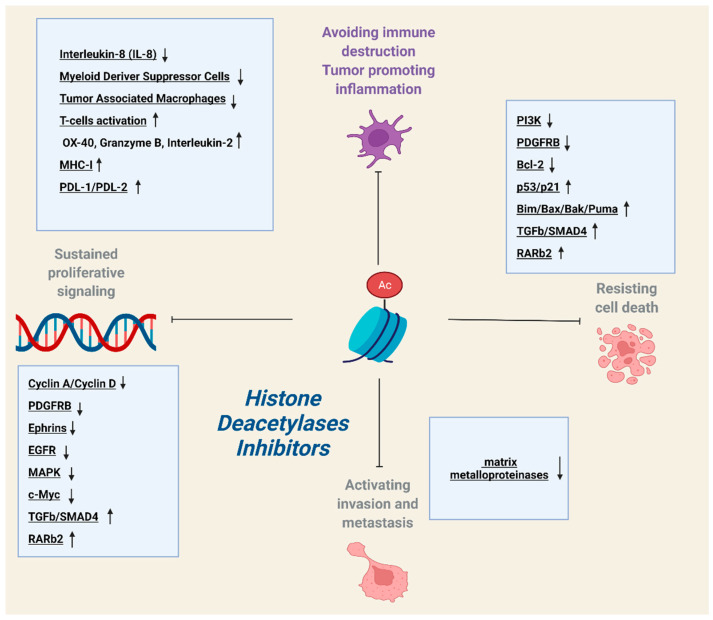
HDAC inhibitors (HDACi) exert pleiotropic effects on different vital cellular processes, such as proliferation, apoptosis, metastasis and immunogenicity, while they also modulate the recruitment, activation state and tumoricidal potential of immune cells within tumor microenvironment (TME).

**Figure 2 ijms-23-03660-f002:**
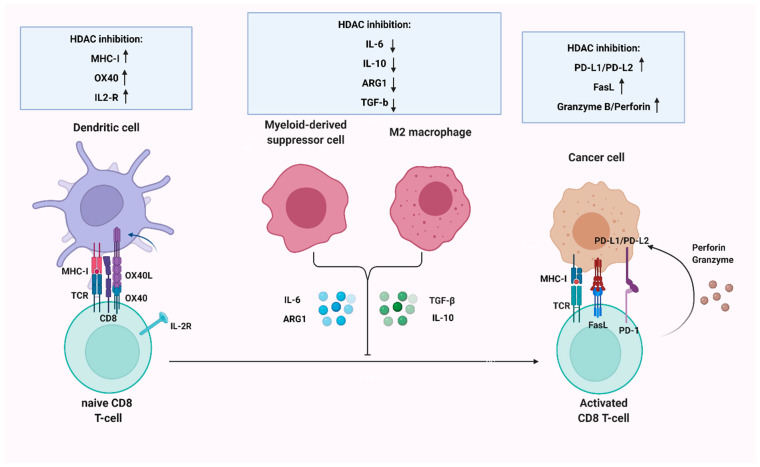
HDAC inhibition interferes with different aspects of immune surveillance by regulating antigen presentation and effective priming of CD8 T-cells, as well as modulating their cytotoxic capacity. It also sensitizes tumors to immune checkpoint inhibitors, by upregulating PD-L1/PD-L2 levels on melanoma cells and alleviates the immunosuppressive functions of myeloid-derived suppressor cells (MDSCs) and tumor-associated macrophages (M2 macrophages).

**Table 1 ijms-23-03660-t001:** Table summarizes the HDACi tested in melanoma in experimental in vitro and in vivo studies (IC50 values have been derived from cell free assays that test the capacity of each molecule to inhibit the activity of HDAC enzymes).

Class	HDAC Inhibitor	HDAC Class Specificity	Target-Mechanism	Experimental Data	IC-50	References
hydroxamic acid	Trichostatin (TSA)	pan	Proliferation suppression via Tle3 inhibition/suppression of xenografts outgrowth	in vitro/ in vivo	1.8 nM	[68]
	Compound (S)-8	Class	Activation of apoptosis/inhibition of invasion through suppression of matrix metalloproteinases	in vitro	-	[70]
	M344	pan	Proliferation suppression via Tle3 inhibition	in vitro	100 nM	[68]
	Suberoylanilide hydroxamic acid (SAHA, vorinostat)	pan	TGFb/smad4 inhibition/suppression of MDSCs recruitment at the tumor microenvironment	in vitro/ in vivo	10 nM	[81,88]
	Suberohydroxamic Acid (SBHA)	pan	Activation of apoptosis through upregulation of Bcl-2 family proapoptotic factors	in vitro	HDAC1: 0.25 μM HDAC3: 0.3 μM	[75]
	Quisinostat (JNJ-26481585)	Class I/II	Upregulation of MHC-I	in vitro	0.11 nM	[83]
	LBH589 (Panobinostat)	Class I/II/IV	Inhibition of PI3K/Akt and c-myc/ upregulation of PDL-1 and PDL-2/activation of T-cells	in vitro/ in vivo	5 nM	[76,94,95]
	MC1568	Class IIa	Interleukin-8/c-Jun suppression	in vitro	100 nM	[82]
	MC1575	Class IIa	Interleukin-8/c-Jun suppression	in vitro	100 nM	[82]
	ACY-1215 (Ricolinostat)	Class IIa	Induction of cell-cycle arrest and activation of apoptosis/suppression of xenografts outgrowth	in vivo	5 nM	[84]
	LAQ824 (Dacinostat)	pan	Reactivation of retinoic acid receptor 2 (RAR2b) expression/activation of apoptosis/suppression of xenografts outgrowth	in vitro/ in vivo	32 nM	[80]
short chain fatty acids	Valproic acid	Class I/IIa	Sensitization of melanoma cells to chemotherapeutic agent temozolomide and radiation therapy/inhibition of DNA double strand breaks repair mechanism/ inhibition of MDSCs tumor suppressive activity/suppression of xenografts outgrowth	in vitro/ in vivo	0.5–2 mM	[73,74,85,89]
	AR42	Class I/II	Downregulation of chaperone proteins and tyrosine kinase pathways (PDGFRB)/sensitization of melanoma cells and xenografts to multikinase inhibitor pazopanib	in vitro	30 nM	[72]
Cyclic peptides	Apicidin	Class I	Tle3 inhibition	in vitro	0.7 nM	[68]
	Depsipeptide	Class I	Upregulation of MHC-II/ upregulation of T-cells effector molecules/sensitization of tumor cells to Fas-FasL mediated apoptotic cell death	in vivo	HDAC1: 36 nM HDAC2: 47 nM	[91,92,93]
benzamides	Entinostat	Class I	Sensitization of melanoma cells to chemotherapeutic agent temozolomide/inhibition of DNA double strand breaks repair mechanism/suppression of xenografts outgrowth	in vitro/in vivo	HDAC1: 0.51 μM HDAC3: 1.7 μM	[85]
sirtuin inhibitors	Tenovin 6	Sirtuins	Activation of apoptosis via upregulation of p53 and proapoptotic factors Bax and Puma/inhibition of invasion via repression of matrix metalloproteinases	in vitro	SIRT1: 10 μM SIRT2: 21 μM SIRT3: 67 μM	[69]
	AC-93253	Sirtuins	Inhibition of EGFR/Ephrins	in vitro	-	[71]
dual inhibitors	Corin (entinostat and lysine demethylase 1 inhibitor tranylcypromine analog)	Class I	Upregulation of p21, mxd1/ Suppression of xenografts outgrowth	in vivo	HDAC1: 0.51 μM HDAC3: 1.7 μM	[86]
other non-classified inhibitors	Ginsenoside Rg3	Class I (HDAC3)	Inhibition of MAPK and PI3K/Akt pathways/upregulation of p53	in vitro	-	[77,79]
	6-Prenylnaringenin (6-PN)	pan	Inhibition of MAPK and PI3K/Akt pathways	in vitro	-	[78]
	8-Prenylnaringenin (8-PN)	pan	Inhibition of MAPK and PI3K/Akt pathways	in vitro	-	[78]
	Nexturastat A	Class IIb (HDAC6)	Phenotypic switch of macrophages from a tumor-promoting (TAM) towards a tumor-suppressive state (M1)	in vivo	5 nM	[90]

**Table 2 ijms-23-03660-t002:** Table summarizes all melanoma clinical trials, either completed or in progress, utilizing HDACi.

Phase	HDAC	Additional Drugs	Condition	Status	NCI Registration Number	References
I	Vorinostat	-	Advanced BRAF V600 mutated melanoma, refractory to BRAF and MEK inhibitors	ongoing	NCT02836548	[96]
I	Panobinostat	Ipilimumab	Unresectable stage III/IV melanoma	completed	NCT02032810	[97]
I	Vorinostat	Marizomib	Melanoma	completed	NCT00667082	[98]
I	Vorinostat	Doxorubicin	Melanoma	completed	NCT00331955	[99]
I	Mocetinostat	Nivolumab, ipilimumab	Unresectable stage III/IV melanoma	terminated	NCT03565406	-
I	Tinostamustine	Nivolumab	Unresectable stage III/IV melanoma	recruiting	NCT03903458	-
I	Panobinostat	-	Unresectable stage III/IV melanoma	completed	NCT01065467	[100]
I	Quisinostat	-	Melanoma	completed	NCT00677105	[101]
I/II	Pivanex	-	Melanoma relapsed after chemotherapy or Interleukin-2 (IL-2) treatment	terminated	NCT000877477	-
I/II	Entinostat	Pembrolizumab	Unresectable or metastatic melanoma resistant to anti PD-1/PDL-1 treatment	ongoing	NCT02437136	[102]
I/II	Valproic acid	Karenitecin	Stage IV melanoma	terminated	NCT00358319	[103]
I/II	Panobinostat	Temozolamide, Decitabine	Unresectable stage III/IV melanoma (cutaneous, ocular, mucosal)	completed	NCT00925132	[104]
I/II	Abexinostat	Pembrolizumab	Unresectable stage III/IV melanoma (cutaneous)	recruiting	NCT03590054	-
II	MS-275	-	Unresectable stage III/IV melanoma (cutaneous, mucosal) resistant to a previous systemic treatment	completed	NCT00185302	[105]
II	Entinostat	Pembrolizumab	Metastatic uveal melanoma	completed	NCT02697630	[106,107]
II	Belinostat	Binimetinib	Metastatic uveal melanoma	recruiting	NCT05170334	-
II	Vorinostat	-	Metastatic uveal melanoma	ongoing	NCT01587352	-
II	Vorinostat	-	Advanced, unresectable or metastatic cutaneous or uveal melanoma	completed	NCT00121225	[92]
II	Entinostat	Pembrolizumab	Unresectable or metastatic melanoma	recruiting	NCT03765229	-

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
