# Peer review of "Histone Deacetylase (HDAC) Inhibitors: A Promising Weapon to Tackle Therapy Resistance in Melanoma"

_ijms, 2022, doi:10.3390/ijms23073660_

Round 1

Reviewer 1 Report

The review entitled « Histone Deacetylases (HDAC) Inhibitors: A Promising Weapon to Tackle Therapy Resistance in Melanoma” is a well-written and informative review about pathogenesis of melanoma and pathways involved in therapy resistance. Moreover, authors describe the use of HDAC inhibitors from bench to clinical trials as an effective strategy to reduce therapy resistance in melanoma.  The following points should be considered:

  • Authors should briefly explain the use of immune checkpoint inhibitors in melanoma (anti-PD1, anti-PD-L1/2, anti-CTLA4…). In the part 2.4 authors well-explained mechanisms of tumor immune escape associated to a reduced surface expression of the MHC-I/peptide. It would be highly interesting to explain other mechanisms of therapy resistance associated to TME.
  • What is the effect of HDAC inhibitors on activity of specific immune population involved in cancer immune tolerance and/or cancer immune-edition? Are there evidences HDAC inhibitors increase expression and presentation of neo-antigens in melanoma. In my point of view, this information will help the reader to understand how HDAC inhibitors can overcome resistance to ICIs.
  • The authors should briefly explain the different populations of MDSCs and TAMs. Do these cells show epigenetic alteration in melanoma? By which mechanisms valproic acid, administered in a xenograft model, induced a phenotypic switch of MDSCs into anti-tumor phenotype? Different examples are given on immune regulation by HDACi, but in many cases only the final effect is written as a fact, without a short explanation what is known about the mechanism.
  • It would be interesting to indicate the IC50 of each HDAC inhibitor in table 1
  • Authors should improve the visibility of Figure 1. The central image showing histone modifications is not visible.
  • Toxicities and adverse effects of HDAC inhibitors in vivo and clinical trials are not discussed in the review
  • The hybrids of HDACis with one or more other cancer target (e.g. HDAC/topoisomerase inhibitors, HDAC/Tyrosine kinase inhibitors) are not developed in the review

Author Response

We would like to thank the reviewer for his/her meaningful comments and suggestions. Our responses can be found below:

  1. We have added a brief analysis about the utilization of immune checkpoint inhibitors in melanoma, in introduction (lines 59-66) and in section 2 (lines 112-116).
  2. Regarding the effects of HDAC inhibition in different components of tumor microenvironemnt, there is more in-depth analysis in the section 2.3. This part of the manuscript focuses on the pleiotropic immunomodulatory activity of HDAC inhibitors in melanoma. Lines 386-402 describe their effect on immunosuppressive myeloid cell populations, such as TAMs and MDSCs. Moreover, a number of mechanisms, concerning interaction among tumor cells and adaptive immunity cells are analyzed: upregulation of MHC-I, Fas/FasL and of T-cell effector molecules, such as granzyme B (lines 414-420). We have also added a paragraph, describing three additional mechanisms: transcriptional upregulation of PDL-1/2, interleukin-2 receptor (IL-2R) and of costimulatory molecule OX-40 (lines 421-430). Moreover we have introduced a figure (Figure 2), that summarizes the effect of specific HDACi to different levels of immune response.
  3. We have added some mechanistic insight, concerning the effect of HDACi in regulating the immunosuppressive potential of myeloid cellular subsets of innate immunity, such as MDSCs and TAMs (lines 386-402).
  4. We have replaced the central image on figure 1, with one with better resolution.
  5. We have added a section about major (≥ Grade 3) side effects and other limitations in the effectiveness of HDACi, as therapeutic approach in melanoma (line 528: “HDACi limitations-adverse effects”).
  6. We have added the IC50 of HDACi in table 1, when available.
  7. Regarding the hybrid molecules, that represent a very promising therapeutic strategy in multiple malignancies. In melanoma, the only hybrid molecule we found to have been tested is Corin, a combination of Entinostat and Lysine demethylase inhibitor, which is analyzed in section 3.1 (lines 344-350).

Reviewer 2 Report

Kostas et al. have summarized current outcomes of clinical trials which used HDAC inhibitor for malignant melanoma. They described the characteristic of melanoma and clinical potential of HDAC inhibitors. The manuscript is well structured and the writing is also good. Furthermore, the tables embedded in the manuscript are well organized and easy to follow. The reviewer believes this manuscript is suitable for IJMS. The reviewer only raises minor point to be addressed.
1. The authors described the current limitation of HDAC inhibitor in melanoma treatment. But the reviewer thinks it should be separately summarized under the title “Limitations” or similar. The reviewer is afraid of readers' misunderstanding that HDACi is almighty for melanoma.

Author Response

We would like to thank the reviewer for his/her meaningful comments and suggestions. As a response, we have added a section “HDACi limitations-adverse effects” (line 528), that focuses on the currently existing restrictions in HDACi implementaton.